# TssA forms a gp6-like ring attached to the type VI secretion sheath

Sara Planamente[1], Osman Salih[2], Eleni Manoli[1], David Albesa-Jové[3,4], Paul S Freemont[2,*] & Alain Filloux[1,**]

## Abstract

The type VI secretion system (T6SS) is a supra-molecular bacterial complex that resembles phage tails. It is a killing machine which fires toxins into target cells upon contraction of its TssBC sheath. Here, we show that TssA1 is a T6SS component forming dode-cameric ring structures whose dimensions match those of the TssBC sheath and which can accommodate the inner Hcp tube. The TssA1 ring complex binds the T6SS sheath and impacts its behaviour *in vivo*. In the phage, the first disc of the gp18 sheath sits on a baseplate wherein gp6 is a dodecameric ring. We found remarkable sequence and structural similarities between TssA1 and gp6 C-termini, and propose that TssA1 could be a baseplate component of the T6SS. Furthermore, we identified similarities between TssK1 and gp8, the former interacting with TssA1 while the latter is found in the outer radius of the gp6 ring. These observations, combined with similarities between TssF and gp6N-terminus or TssG and gp53, lead us to propose a comparative model between the phage baseplate and the T6SS.

**Keywords** bacteriophage baseplate; gp6; T6SS; TssA; type VI secretion system
**Subject Categories** Microbiology, Virology & Host Pathogen Interaction; Structural Biology
**The EMBO Journal (2016) 35: 1613–1627**

## Introduction

Bacteria face fierce competition for resources in their environment or hosts. A molecular machine, called the type VI secretion system (T6SS), has emerged as a weapon (Filloux, 2013) that delivers toxins directly into bacterial competitors (Russell *et al*, 2011). For example, the peptidoglycan hydrolases Tse1 and Tse3 in *Pseudomonas aeruginosa* (Russell *et al*, 2011), or the nucleases Tde1 and Tde2 in *Agrobacterium tumefaciens* (Ma *et al*, 2014),

challenge the integrity of the cell wall or the genetic material (Benz & Meinhart, 2014). T6SS-dependent bacterial confrontation plays a role in the persistence of *P. aeruginosa* in the lungs of cystic fibrosis patients (Mougous *et al*, 2006), or the establishment of *Bacteroidetes* as members of the gut microbiota (Russell *et al*, 2014). Several T6SS effectors also manipulate eukaryotic cells (Hachani *et al*, 2016). For example, VgrG1 in *Vibrio cholerae* (Pukatzki *et al*, 2007), or VgrG2b in *P. aeruginosa* (Sana *et al*, 2015), modulates internalization of the bacterium by interacting with the actin or tubulin network, respectively. T6SS effectors can also act on the membrane of prokaryotic and eukaryotic cells, for example the PldA and PldB (Jiang *et al*, 2014) phospholipases produced by *P. aeruginosa*. The T6SS thus has a crucial role in mediating interactions between living cells, but its structure and mode of action has yet to be fully elucidated.

Protein secretion systems span the cell envelope of Gram-negative bacteria with an inner membrane (IM) complex connected to another embedded in the outer membrane (OM) (Filloux, 2011). Usually, the OM complex forms a pore through which toxins or enzymes are released. The T6SS is unique in that it has no obvious open OM complex but instead uses a phage tail spike-like structure (Hcp/VgrG) to puncture the cell envelope (Leiman *et al*, 2009). The tail docks into a membrane complex formed by two integral IM proteins (TssL/TssM) and one lipoprotein (TssJ) (Durand *et al*, 2015). The puncturing induces conformational changes in the otherwise closed membrane complex, allowing extrusion of T6SS toxins (Durand *et al*, 2015). Such puncturing by the Hcp/VgrG tail spike relies on the assembly of a contractile sheath, namely TssB/TssC, or VipA/VipB in *V. cholerae* (Lossi *et al*, 2013; Kube *et al*, 2014; Kudryashev *et al*, 2015), which is connected to the inner face of the IM via a putative baseplate (Basler *et al*, 2012). The T6SS baseplate concept is an extrapolation of what is extensively described in phages (Leiman *et al*, 2010). Because of an ancient divergence between the phage tail and the T6SS, limited sequence homologies exist between these structures. However, a number of T6SS proteins of unknown function are good candidates for being part of a baseplate-like structure.

1  MRC Centre for Molecular Bacteriology and Infection (CMBI), Department of Life Sciences, Imperial College London, London, UK
2  Section of Structural Biology, Department of Medicine, Imperial College London, London, UK
3  Unidad de Biofísica, Departamento de Bioquímica, Consejo Superior de Investigaciones Científicas - Universidad del País Vasco/Euskal Herriko Unibertsitatea (CSIC-UPV/EHU), Universidad del País Vasco, Leioa, Bizkaia, Spain
4  Structural Biology Unit, CIC bioGUNE, Bizkaia Technology Park, Derio, Spain
   *Corresponding author. Tel: +44 20 7594 5327; E-mail: p.freemont@imperial.ac.uk
   **Corresponding author. Tel: +44 20 7594 9651; E-mail: a.filloux@imperial.ac.uk

Here, we show that TssA1 from *P. aeruginosa* is an essential T6SS component and forms a ring-shaped complex positioned at one end of the TssB/TssC sheath. The dimension of the TssA1 ring is similar to the sheath diameter and we confirmed the proximity with the T6SS tail-like structure by showing interaction between TssA1 and Hcp1. We highlight a role for TssA1 in the assembly of the TssB1C1 sheath *in vivo* and identify remarkable sequence and structural homologies between TssA1 and the phage baseplate protein gp6. We propose that TssA1 could be a T6SS baseplate-like component that connects the sheath structure to the TssJLM membrane complex via an interaction with the TssK1 protein. TssK has been shown to contact TssL (Zoued *et al*, 2013) and we have further expanded the structural similarity between the T6SS and phage baseplates by suggesting that TssK could possibly be a gp8 homologue.

## Results

### TssA1 is an essential multimeric component of the T6SS

In *P. aeruginosa,* there are three T6SS clusters (H1- to H3-T6SS) (Filloux *et al*, 2008), each encoding all proteins that form the T6SS core (Boyer *et al*, 2009). The genetic organization of these clusters and the level of amino acid conservation between T6SS components vary which suggests that they are not the result of gene duplication events. TssA is a conserved T6SS core component, but phylogenetic analysis showed that TssA1, TssA2 and TssA3, encoded in the H1-, H2- and H3-T6SS clusters, respectively, are found in distinct branches (Fig 1A). Here, we studied H1-T6SS (Fig 1B) and assessed the function of TssA1 *in vivo*. A *tssA1* deletion was engineered, which abolished the secretion of Hcp1, Tse3 and VgrG proteins, all markers of *P. aeruginosa* T6SS activity (Fig 1C), thus confirming the essential role of TssA1.

TssA1 is a 344 amino acids protein (molecular weight (MW) ~37 kDa) with no predicted signal peptide or transmembrane domain. Size-exclusion chromatography (SEC) shows that the purified N-terminally His$_6$-tagged TssA1 (His$_6$-TssA1, MW ~40 kDa) elutes at an early elution volume, suggesting that TssA1 oligomerizes in solution. The propensity of TssA1 to self-associate was confirmed *in vivo* using bacterial two-hybrid (BTH) assays (Fig 2A) and *in vitro* by chemical cross-linking experiments (Fig 2B). The cross-linked TssA1 shows a series of species consistent with dimeric (~80 kDa), trimeric (~120 kDa) and hexameric (~240 kDa) forms, together with a band corresponding to a MW higher than 315 kDa. These data suggest that TssA1 protomers could assemble to form higher-order complexes greater than hexamers (Fig 2B). To obtain a more accurate estimate of the oligomeric state of TssA1, we carried

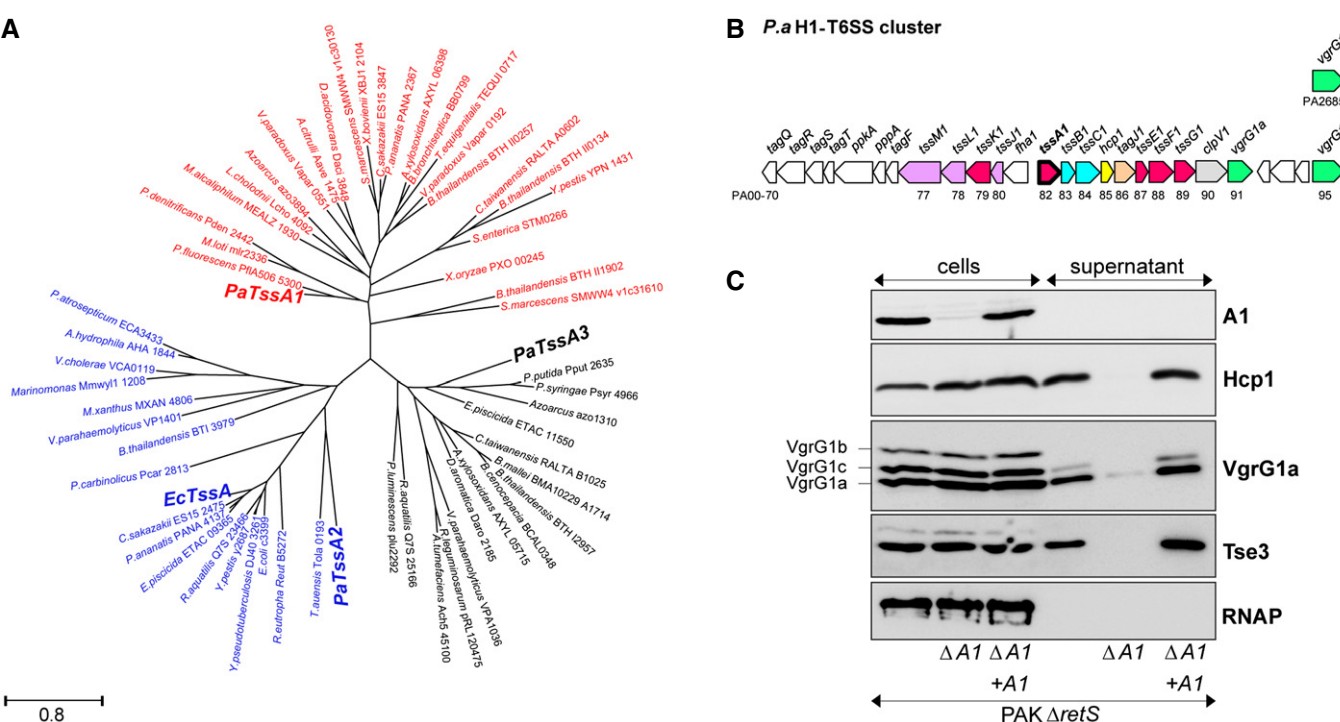

**Figure 1. TssA1 is an essential component of the T6SS apparatus.**

A Maximum-likelihood phylogenetic tree generated from 61 aligned TssA sequences belonging to the indicated bacterial species. The three TssA proteins from *P. aeruginosa* (PaTssA1, PaTssA2 and PaTssA3) and *E. coli* TssA (EcTssA, GenBank accession number: 284924261) are indicated in bold.

B Graphical representation of the H1-T6SS gene cluster in *P. aeruginosa* (*P.a*) (Filloux *et al*, 2008). The generic *tss* name and the PA number of corresponding genes are indicated. Genes encoding putative baseplate components are shown in magenta with *tssA1* highlighted in bold.

C Proteins from whole-cell extracts and culture supernatants of PAKΔ*retS*, the derivative *tssA1* mutant (Δ*A1*) (both carrying the pBBR plasmid) and the complemented Δ*A1* mutant carrying pBBR-*tssA1* (Δ*A1 + A1*) were analysed by Western blot. Polyclonal antibodies directed against TssA1 (A1), Hcp1, VgrG1a and Tse3 were used. The anti-VgrG1a antibody detects VgrG1a, VgrG1b and VgrG1c, as indicated on the left. Cytoplasmic RNA polymerase (RNAP) was monitored using monoclonal antibody directed against its β-subunit.

out analytical ultracentrifugation (AUC) experiments. Sedimentation velocity data were collected for His$_6$-TssA1 at concentrations between 0.32 and 0.8 mg/ml, at two rotor speeds 20,000 and 30,000 rpm, and a representative experiment is shown in Fig 2C. The size-distribution analyses c(s) reveal a major dodecamer peak consistently observed at 15.8–16.6 S with a molecular mass of 479 ± 5 kDa (Fig 2C). A minor peak is also observed at 28.3–29 S which yields a molecular mass of 1.05 ± 0.05 MDa. SEC coupled with multiangle light scattering (SEC-MALS) measurements also indicated that His$_6$-TssA1 exists in different oligomeric states in solution (Fig EV1A). The first oligomeric state that could be resolved has a molar mass of 1.0 ± 0.1 MDa which is consistent with the double-dodecamer (24-mer) TssA1 complex seen by AUC experiments. However, the oligomeric state of the following peaks could not be evaluated by SEC-MALS due to peak overlap (Fig EV1A).

Altogether, these data are consistent with TssA1 being predominantly a dodecamer in solution, although its ability to form a double-dodecamer complex cannot be excluded.

### TssA1 forms ring-shaped structures

To study the quaternary structure of TssA1, we used negative stain electron microscopy (EM). TssA1 complexes appear as distinct ring-shaped structures (Fig 3A) with lobes distributed around a central hole. The rings have an average diameter of ~260 Å, while the central hole measures ~100 Å across. Isolated, mono-disperse rings that have high contrast are clearly visible and exhibit discrete features so that single-particle approach could be used. A data set of ~4,000 TssA1 rings was selected from the micrographs and two independent image processing approaches were used to analyse the

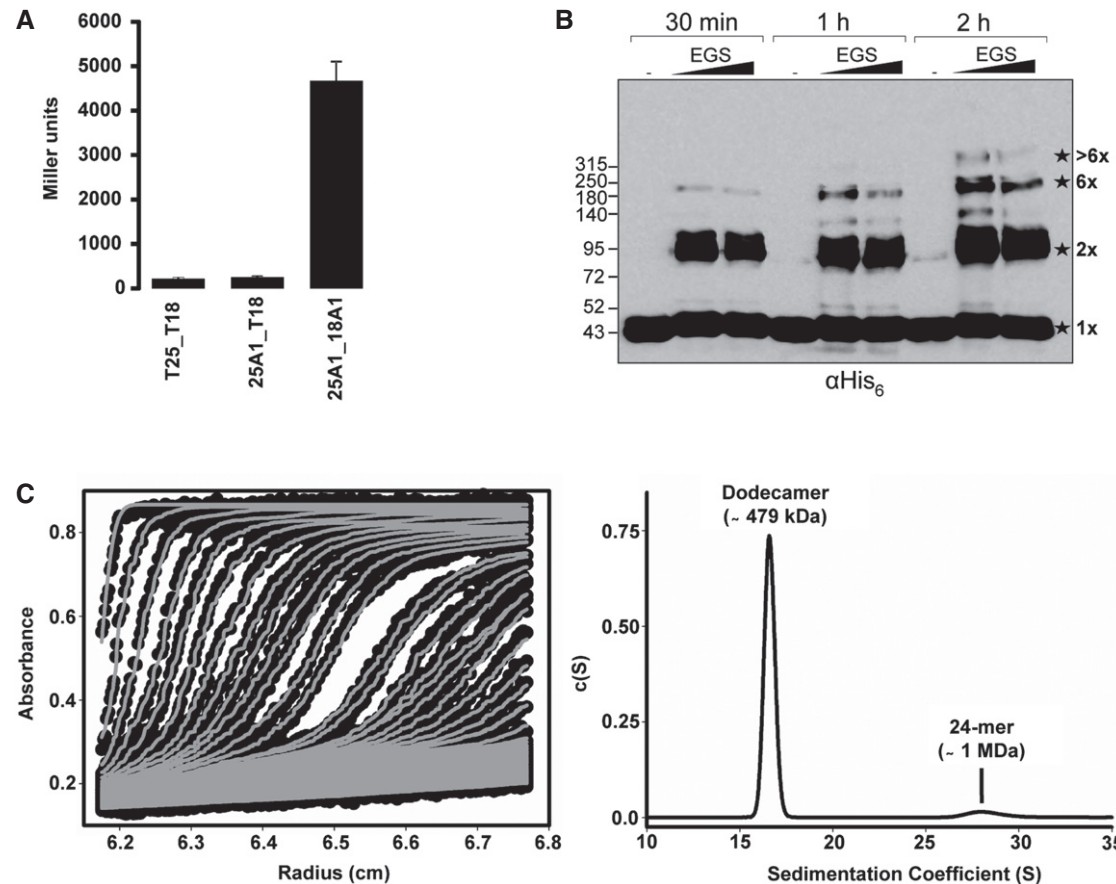

**Figure 2. TssA1 forms a homo-multimeric complex.**

A   BTH experiments showing that TssA1 is able to self-interact *in vivo*. A graphical representation of β-galactosidase activity from co-transformants of *E. coli* DHM1 cells producing TssA1 (A1) fused to the adenylate cyclase T25 or T18 subunits is shown. The values shown on the *y*-axis correspond to the activity in Miller units. In each case, average activity from two independent samples is shown and error bars indicate the standard deviation (SD). Experiments were carried out in quadruplicate.

B   *In vitro* cross-linking experiments of purified His$_6$-TssA1. About 30 μg of purified His$_6$-TssA1 was cross-linked (30 min, 1 and 2 h) at room temperature using increasing amounts of ethylene glycol-bis(succinimidylsuccinate) (EGS; 2 and 5 mM) where indicated. Western blot analysis of cross-linked products using an anti-His$_6$ monoclonal antibody is shown. The cross-linked species are highlighted with stars with the corresponding oligomeric state indicated on the right (1× = monomer; 2× = dimer; 6× = hexamer). Molecular weight markers (kDa) are indicated on the left.

C   Analysis of the oligomeric state of TssA1 by AUC. Sedimentation data of His$_6$-TssA1 (0.56 mg/ml) recorded at a rotor speed of 20,000 rpm is shown. The left panel shows the sedimentation boundary fits and for clarity, only every third scan is shown in the fitted data plots. The experimental absorbance data are shown as black circles, whereas the boundary fits are shown as grey lines. The right panel shows the size-distribution analysis c(s), obtained from fitting the scan boundaries using SEDFIT, revealing a dodecameric peak (Mr ~479 kDa) at S$_{20,w}$ value of 16.6 S with a minor peak (Mr ~1 MDa) at S$_{20,w}$ value of 29.01 S.

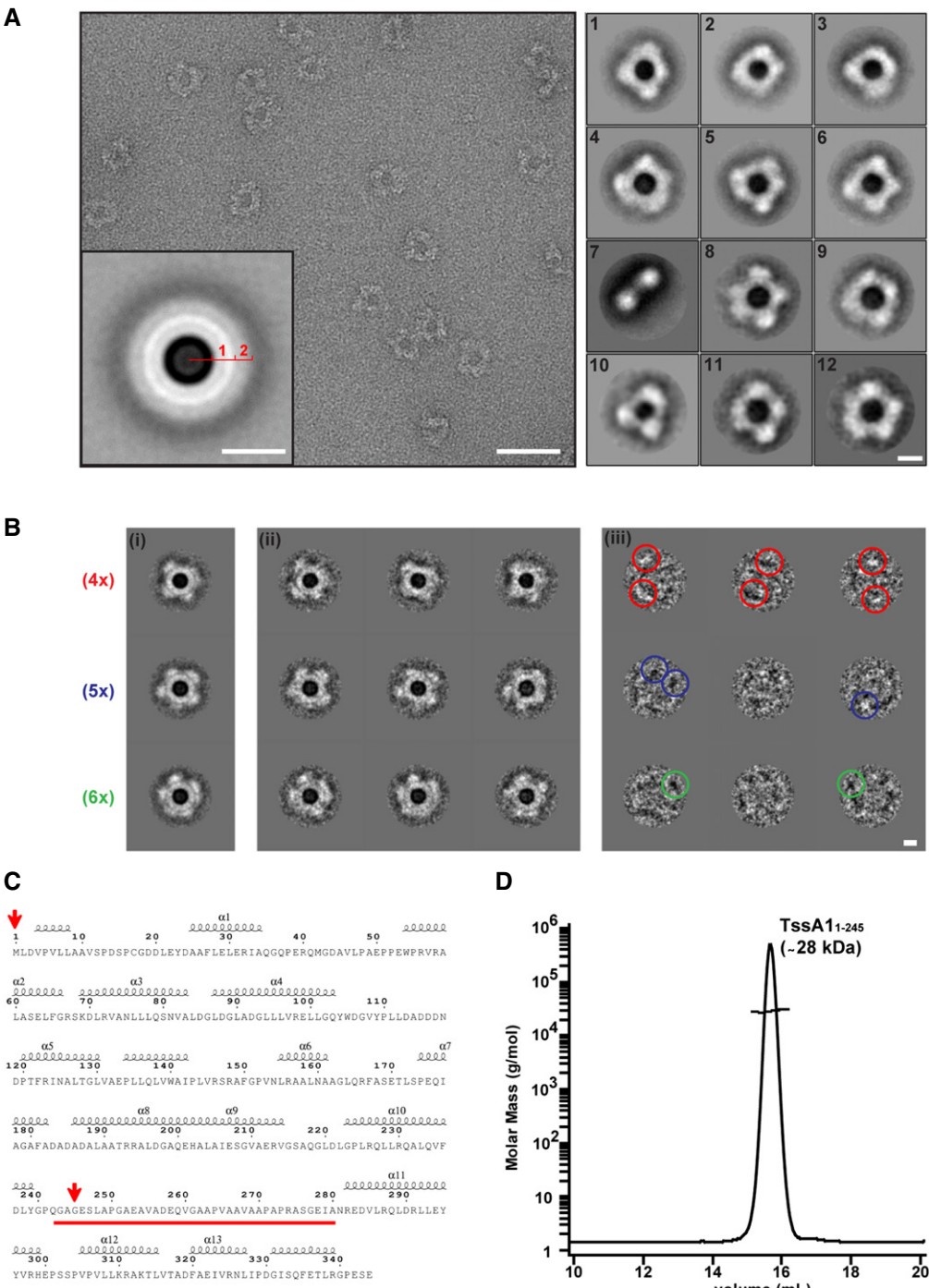

**Figure 3. TssA1 forms ring-shaped structures.**

A   A representative micrograph of the TssA1 complex (left panel) showing discrete ring-shaped particles with a diameter of ~260 Å. The rotationally symmetrized average of the data set (inset panel) presents two distinct rings (numbered), disclosing the layered nature of the TssA1 complex. Representative class averages of the TssA1 complex (right panel) demonstrating the wide range of conformations taken by the ring-shaped structures. Scale bars are 500 Å in the left panel, 130 Å in the inset and 100 Å in the right panels, respectively.

B   Double MSA classification: (i) representative orientation class averages show TssA1 rings arranged into fourfold (4×), fivefold (5×) and sixfold (6×) symmetry-like complexes (indicated in red, blue and green, respectively); (ii) subclass averages obtained from subclassification of the corresponding orientation class; and (iii) calculated difference images showing some white or black peaks (coloured circles) at the outer ring region. Scale bar is 100 Å.

C   Secondary structure prediction based on the amino acid sequence of TssA1 using the web-based prediction server ESPript 3 (Robert & Gouet, 2014). The linker region from Gly[240] to Ala[280] is highlighted in red. The positions of the first and last residues of the TssA1[1–245] truncated protein are indicated by red arrows.

D   SEC-MALS analysis of the purified His6-TssA1[1–245]. The TssA1[1–245] elution peak is shown and the corresponding mass is indicated. The molar mass and the differential refractive index (RI) are plotted versus the elution volume. Data were analysed using the ASTRA method (Slotboom et al, 2008).

particles. The images were filtered, normalized, centred and then subjected to reference-free successive multivariate statistical analysis (MSA) steps for classification (van Heel, 1984; van Heel *et al*, 2000; Elad *et al*, 2008) using IMAGIC-5 (van Heel *et al*, 1996) (Fig EV2A). In parallel, the same data set was masked and treated to an empirical Bayesian approach to classification utilized in RELION (Scheres, 2012) (Fig 3A, right panel). The results from both methodologies give similar class averages that show TssA1 oligomers as distinct rings exhibiting extensive conformational heterogeneity in external lobe positions (Fig 3A, right panel and 3B). The rotationally symmetrized average of all TssA1 particles (Fig 3A, inset panel) showed an inner core (ring 1) measuring ~50 Å width and a flexible outer region (ring 2) with ~60 Å width. The 2D class averages and rotationally symmetrized average (Fig 3A, right and inset panels) show that the dimensions of the central hole and the inner core are conserved amongst all particles and that TssA1 particles preferentially assume top-view orientations with very few side views present (~1%) (class 7, Fig 3A, right panel). The outer region (Fig 3A, inset panel) varies between class averages which we attribute to flexibility of the external lobes. The molecular views of the TssA1 complex range from threefold symmetry-like (class 10, Fig 3A, right panel) composed of three large lobes to sixfold symmetry-like (class 12, Fig 3A, right panel) with six smaller, less defined lobes. The variation in lobe size could be due to an uneven distribution of the external part of individual TssA1 protomers within the dodecameric ring which could also account for the uneven spacing between lobes (Fig 3A, right panel).

In an attempt to obtain more homogenous subsets of images, the double MSA classification procedure (Elad *et al*, 2008) was performed to separate between variance due to orientation and variance as a result of local conformational changes (Fig 3B). The first MSA based on low-order eigenimages that correspond to large-scale image variations sorts the data according to orientation and results in as four-, five- and sixfold symmetry-like complexes for TssA1 (Fig 3B). The second MSA based on high-order eigenimages that correspond to fine image variations sorts the data according to attributable local organizational differences. The resultant class averages from the subclassifications were used to calculate difference images to assess the quality of the separation and the variation in lobe position (Fig 3B, right panel). This revealed a few prominent positive or negative peaks (Fig 3B, right panel, coloured circles) at the outer ring region demonstrating some in-plane rotational alignment heterogeneity which prevents unambiguously sorting of particles into separate homogenous subsets. This analysis highlights the flexible nature of the TssA1 lobed-ring complex.

The predicted secondary structure of TssA1 shows the presence of a long disordered alanine-/proline-rich region of 40 amino acids (linker region) which appears to delineate two domains (Fig 3C, highlighted in red). The presence of this linker region is consistent with the conformational flexibility of the TssA1 ring complex. To gain further insights into the organization of the ring complex, the TssA1 C-terminal region (Ser$^{246}$ to Glu$^{344}$) was removed (Fig 3C, red arrows) and the truncated protein purified and analysed by SEC-MALS. As shown in Fig 3D, the average molar mass of the truncated TssA1$_{1-245}$ across the elution peak is $28 \pm 1$ kDa which is the expected molar mass for a monomer. The monomeric state of TssA1$_{1-245}$ was further confirmed by the absence of cross-linked species (Fig EV1B) and by the AUC experiments showing a single

peak with a corresponding mass of $25.6 \pm 0.6$ kDa (Fig EV1C). Overall, these findings suggest that the TssA1 C-terminal domain is likely required for homo-oligomerization and ring formation.

## The TssA1 ring is at one end of the T6SS sheath

In order to identify TssA1 interacting partners within the T6SS complex, we performed pull-down experiments using the *tssA1 P. aeruginosa* mutant expressing His$_6$-tagged TssA1. The resulting strain ($\Delta A1 + His_6$-$A1$) has a functional T6SS (Appendix Fig S1) showing that the N-terminal His$_6$-tag does not affect TssA1 function *in vivo*. Soluble fractions were used for Ni-NTA affinity chromatography and eluted samples were analysed by mass spectrometry (MS). The detected proteins, absent from the control (no His$_6$-tagged TssA1), are listed in Table 1 and include TssB1, a component of the T6SS sheath.

To further assess whether TssA1 interacts with the T6SS sheath complex, we produced His$_6$-TssA1 and untagged TssB1C1 in *Escherichia coli* (Appendix Tables S1 and S2) and carried out Ni-NTA affinity chromatography. As shown in Fig 4, TssB1 (18 kDa) and TssC1 (55 kDa) are co-purified with His$_6$-TssA1. Further, SEC allowed the purification of a stable complex (Fig 4B) that was used for immunogold labelling of His$_6$-TssA1. After removal of unbound gold nanoparticles, the labelled complex was visualized by negative stain EM. The collected electron micrographs show that most of the TssB1C1 sheaths, that is 87 out of the 107 tubules analysed, display one or more gold particles at one extremity (Fig 4C), indicating the presence of His$_6$-TssA1 at one end of the TssB1C1 sheath.

Taken together, our data provide evidence that the TssA1 ring is positioned at one extremity of the TssB1C1 sheath. This is a striking observation given that the TssA1 ring dimensions (~260 Å × ~100 Å) are comparable with those of the TssB1C1 sheath (~250–330 Å × ~100 Å) (Lossi *et al*, 2013) and consistent with the conserved sheath dimensions in bacteria and bacteriophage (~240–330 Å × ~95–120 Å) (Clemens *et al*, 2015; Kostyuchenko *et al*, 2005; Kudryashev *et al*, 2015; Leiman *et al*, 2010, 2004). It has been proposed that Hcp rings (90 Å diameter) (Mougous *et al*, 2006) fit within the cavity of the TssBC sheath and we would propose that Hcp hexamers also fit the inner cavity of the TssA1 ring as supported by BTH experiments that show TssA1 and Hcp1 interact (Fig 4D).

In summary, these results underline the compatibility between the newly discovered TssA1 ring and the TssB1C1–Hcp1 complex.

**Table 1.   MS data showing T6SS components present in the His$_6$-TssA pull-down sample.**

| Accession | Identified proteins | PLGS score | Coverage (%) | Precursor RMS mass error (ppm) |
|---|---|---|---|---|
| **PA0082** | **TssA1** | **91,877.8** | **83.4** | **2.1** |
| PA0079 | TssK1 | 1,684 | 41.2 | 6.9 |
| PA0085 | Hcp1 | 621.8 | 46.3 | 7.3 |
| PA0087 | TssF1 | 467 | 30.4 | 11.9 |
| PA0090 | ClpV1 | 428.3 | 26.1 | 5.4 |
| PA0083 | TssB1 | 421.6 | 100 | 17.3 |

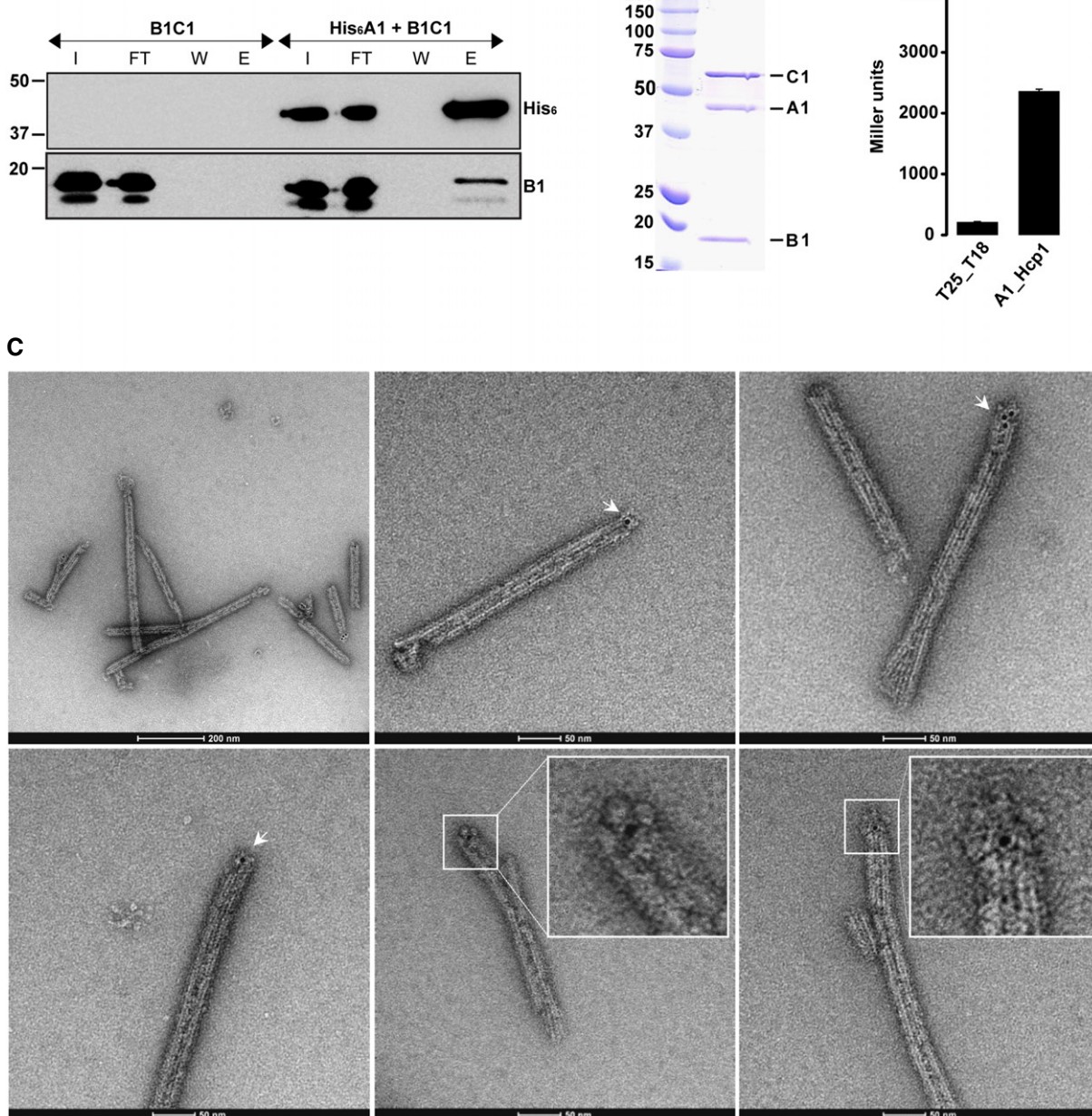

**Figure 4. TssA binds at one end of TssBC tubules.**

A   Co-purification experiments using His₆-TssA1 and untagged TssB1C1. Western blot analysis of the Ni-NTA outputs using antibodies directed against the His₆-tag (upper panel) and TssB1 (lower panel). I, input; FT, flow through; W, wash; E, elution.

B   SEC elution fraction of the co-purified His₆-TssA1 and TssB1C1. SDS–PAGE showing the presence of the three proteins His₆-TssA1 (A1), TssB1 (B1) and TssC1 (C1). Molecular weight markers (kDa) are indicated on the left.

C   Representative micrographs of the immunogold-labelled complex His₆-TssA1/TssB1C1. The presence of gold particles is indicated with white arrows. Two close-up views of TssB1C1 sheath displaying gold particles at one extremity are shown (inset panels). Scale bars are 2,000 and 500 Å for images 1 and 2–6, respectively.

D   BTH experiment showing interaction between TssA1 (A1) and Hcp1. A graphical representation of β-galactosidase activity from *E. coli* DHM1 cells producing the indicated proteins fused to the adenylate cyclase T25 or T18 subunits is shown. The values shown on the *y*-axis correspond to the activity in Miller units. In each case, average activity from two independent experiments is shown and error bars indicate the standard deviation (SD). Experiments were carried out in triplicate.

## Structural similarities between TssA1 and the C-terminal moiety of gp6

There is a striking parallel between the quaternary arrangement and size of the TssA1 ring complex and the gp6 complex from the T4

phage baseplate (Aksyuk *et al*, 2009b). For gp6, 12 copies of the C-terminal region (gp6_334C, residues 334–660, MW 37 kDa) assemble to form a central ring within the baseplate (PDB codes 3H3W and 3H3Y). Furthermore, our analysis of the gp6 ring structure in re-projection images reveals that the dimensions of this

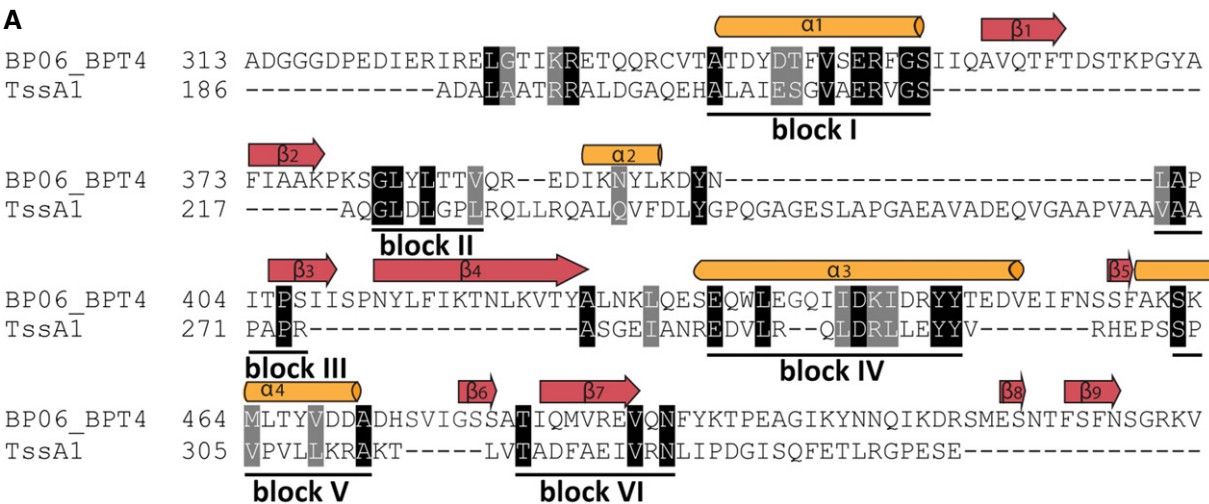

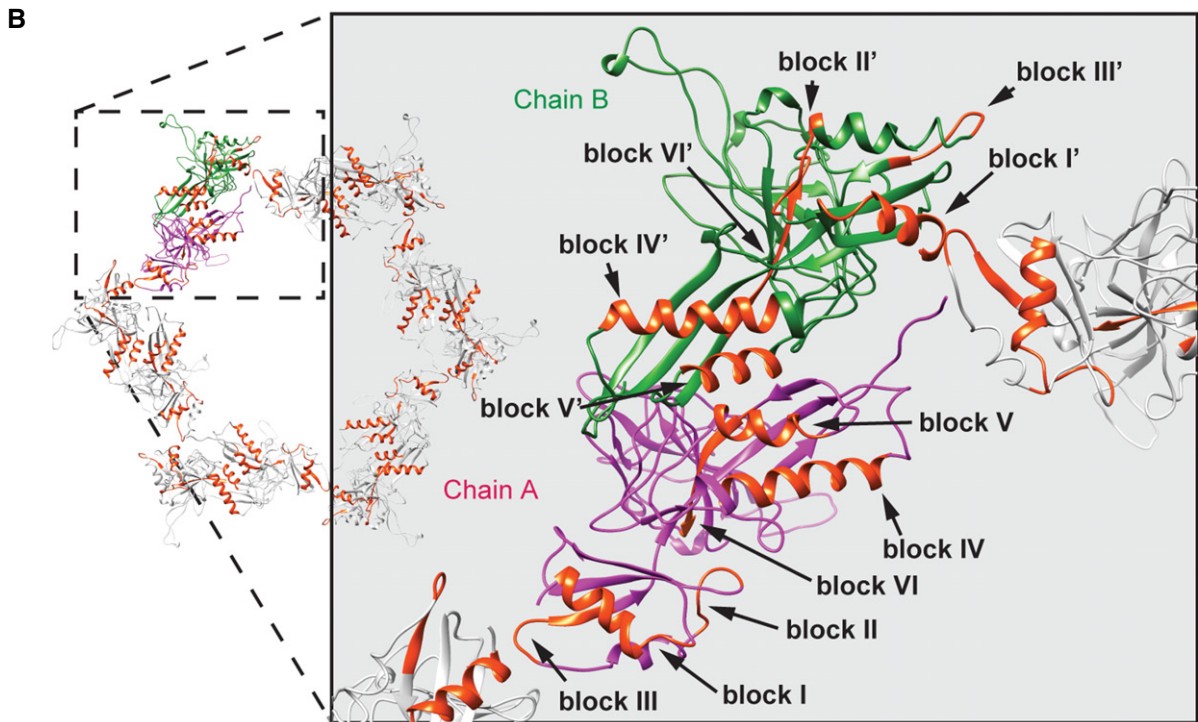

**Figure 5. Secondary-structure-weighted sequence alignment of TssA1 with gp6.**

A   Protein sequences were extracted from UniProt accession number BP06_BPT4 for enterobacteria phage T4 (gp6; bacteriophage T4) and the *Pseudomonas* Genome Database (TssA1; reference strain PAO1). Conserved positions are shown in black and grey background. The secondary structural elements corresponding to the 3D structure of gp6 (PDB code 3H2T) are shown above the alignment. Regions with significant sequence conservation are indicated with a corresponding block number.

B   Cartoon representations of the dodecameric gp6_334C structure (PDB code 3H3W). Chains A and B of gp6 dimers are shown in magenta and green, respectively, and the conserved blocks mapped onto the assembled dodecameric gp6 structure are shown in orange. A close-up view showing the conserved blocks present in regions of gp6 involved in ring assembly is shown on the right.

baseplate component of the T4 phage are consistent with the dimensions of the TssA1 ring (Fig EV2B). To reveal additional sequence and structural similarities, a secondary-structure-weighted sequence alignment of TssA1 with gp6 was carried out using PROMALS3D (Pei *et al*, 2008) and UCSF Chimera (Pettersen *et al*, 2004). We

observe a significant sequence similarity (~62%) within the C-terminal regions of TssA1 and gp6 which clusters to six blocks that can be mapped onto the gp6 structure (Fig 5A). In gp6, some of these blocks are directly involved in assembly of the dodecameric ring structure (Aksyuk *et al*, 2009b) (e.g. block I' (chain B) and block III

(chain A)), whereas other regions mediate dimerization (Fig 5B). These observations support our suggestion that the C-terminal region of TssA1 could mediate oligomerization and ring formation and points at TssA1 as a putative gp6-like baseplate component.

Since TssA1 shows sequence similarity with the C-terminal region of gp6 only, we investigated whether additional T6SS components have sequence similarity with the gp6N-terminus. We found homology between the N-terminal regions of TssF1 and gp6 (12% identity and 63% similarity; Fig EV3A), which confirms previous observation made with TssF of enteroaggregative *E. coli* (Brunet *et al*, 2015). However, a lack of structural information precludes any detailed modelling.

### Does TssA1 impact TssB1C1 sheath assembly?

We next investigated how TssA1 contributes to T6SS sheath assembly *in vivo*. A *P. aeruginosa* mutant deleted for *tssB1* (ΔB1) was engineered and used to produce a TssB1 chimera N-terminally fused with the superfolder GFP (B1-sfGFP). The resulting ΔB1::B1-sfGFP strain and derivative *tssA1* mutant, ΔB1ΔA1::B1-sfGFP, were analysed by fluorescence microscopy. We also analysed another T6SS component, TssE1, using the derivative *tssE1* mutant, ΔB1ΔE1::B1-sfGFP. It has been suggested that TssE is a gp25-like protein (Lossi *et al*, 2011) that forms part of the T6SS basal platform and initiates sheath assembly (Kudryashev *et al*, 2015).

The majority of the observed "parental" ΔB1::B1-sfGFP cells (~79%) displayed fluorescent foci which, as described before (Basler *et al*, 2012), correspond to assembled TssBC sheaths (Fig 6). In marked contrast, no discernable foci were detected in the *tssE1* mutant indicating that assembly of the TssB1C1 sheath does not occur in the absence of TssE1, confirming data from *V. cholerae* (Basler *et al*, 2012). Sheath assembly in the *tssE1* mutant could be complemented by introducing the *tssE1* allele *in trans* (Fig 6). For the *tssA1* mutant, the number of cells displaying foci is still significant (~29%), which indicates that the sheath can be formed in the absence of TssA1 although at much reduced levels (Fig 6). It is noticeable that within *tssA1* mutant cells that have no foci, we observe an increase in the number of cells displaying diffuse fluorescence (~14%) compared to that of other strains where diffuse fluorescence varies between 2 and 8% (Fig 6B). The *tssA1* mutation can also be complemented (Fig 6).

Overall, these results demonstrate that in contrast to TssE1, a lack of TssA1 still allows for sheath formation. It is possible that TssA1 acts downstream of TssE1 to properly attach the pre-assembled sheath onto the baseplate and/or stabilize the sheaths tubular structure.

### TssA1 connections with other T6SS components

Our data suggest that TssA1 could be a T6SS baseplate component that connects the tail-like structure (TssBC/Hcp) with the rest of the machinery. Our pull-down experiments indicate that TssA1 interacts with two other components of the T6SS, namely TssK1 and TssF1 (Table 1), which we further assessed using a BTH approach. The TssA1–TssF1 interaction yields significant β-galactosidase activity (575 ± 50 Miller units; Fig 7), which supports our sequence analysis suggesting that gp6 is a TssF–TssA fusion. More strikingly, TssA1–TssK1 interaction was ~20-fold above control. Previous work

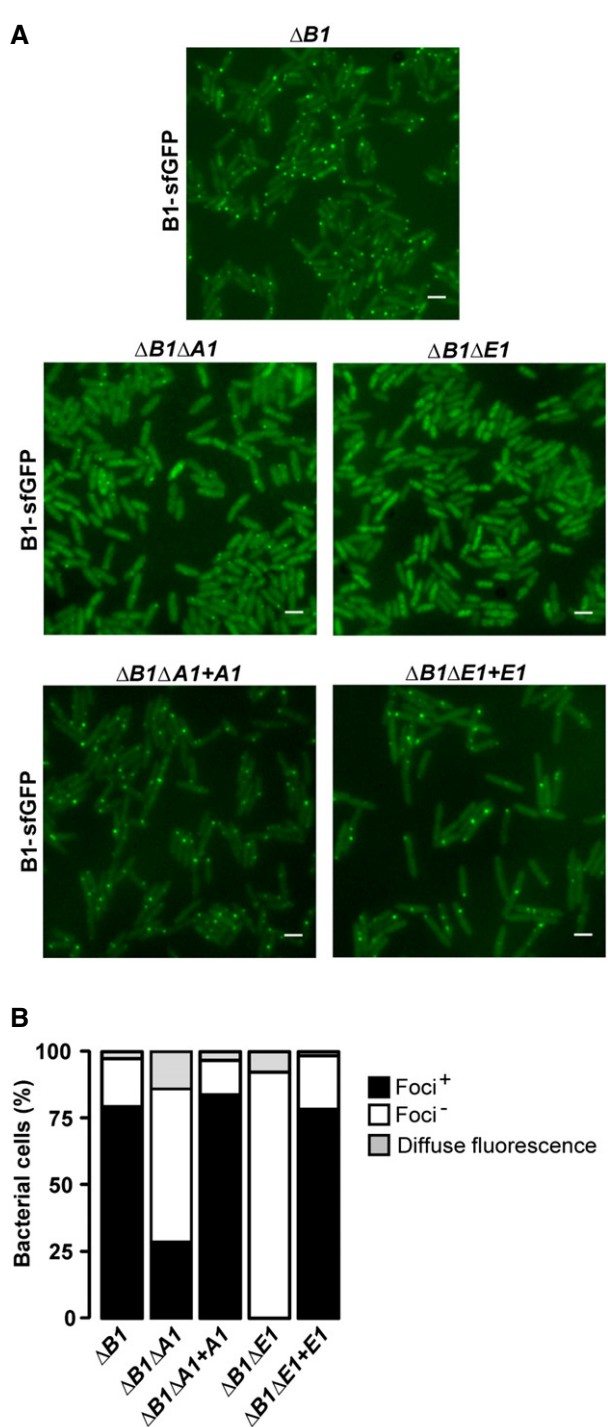

**Figure 6.  TssA1 impacts TssB1C1 sheath assembly *in vivo*.**

A  Fluorescence microscopy images showing the presence/absence of TssBC sheaths (seen as fluorescent foci) in the ΔB1::B1-sfGFP (top panel), ΔB1ΔA1::B1-sfGFP, ΔB1ΔE1::B1-sfGFP cells (middle panels) and in the complemented strains ΔB1ΔA1::B1-sfGFP + A1 and ΔB1ΔE1::B1-sfGFP + E1 (bottom panels). Scale bars are 2 μm.

B  Quantification of TssB1-sfGFP foci and of diffuse fluorescence in various *P. aeruginosa* backgrounds imaged in (A). Values are normalized to the total number of bacteria (%). Data were obtained from three independent experiments, with images and values from representative replicates being shown.

identified TssK as a multimeric component that interacts with the cytoplasmic domains of TssL and TssM (Zoued *et al*, 2013), which suggests that TssK1 connects TssA1 and the T6SS inner membrane complex TssJLM (Durand *et al*, 2015). We found that TssK1 shows some sequence homology to gp8 (similarity 63%, identity 11%; residues 37–328; Fig 8A), with the homology mapping to gp8 regions involved in dimerization and interaction with gp6 (Fig 8B and Appendix Fig S2). Our BTH approach did not detect interactions between TssA1 and TssE1 or TssG1 which are other potential baseplate candidates (Fig 7), but instead an interaction with VgrG1a (Fig 7). VgrG proteins are gp5-gp27 homologues with the latter

described as the central hub in the phage baseplate (Yap *et al*, 2016). Our observation thus strengthens the concept of TssA1 being part of a larger T6SS complex which may display functional similarities with the phage baseplate.

Finally, our pull-down and BTH experiments suggested an interaction between TssA1 and ClpV1 (Table 1 and Fig 7). ClpV is the AAA+ ATPase responsible for TssBC sheath-like structure disassembly (Bonemann *et al*, 2009; Kapitein *et al*, 2013; Forster *et al*, 2014), and our observation suggests that ClpV may not only be involved in TssBC sheath disassembly, but also in recycling other T6SS subcomplexes, such as the TssA ring.

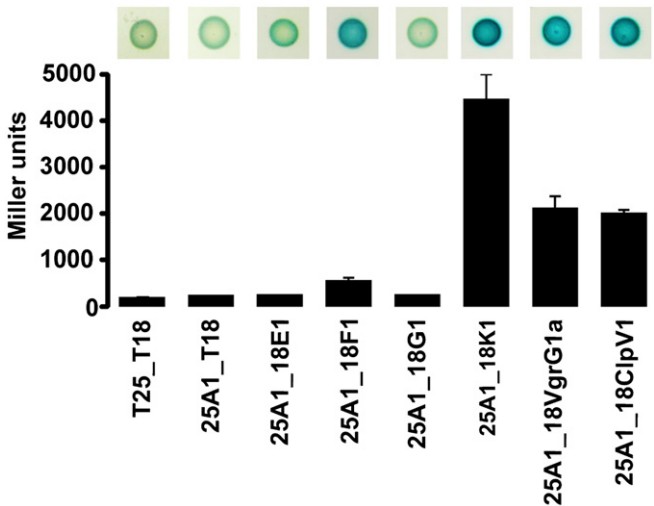

**Figure 7.   The TssA1 interaction network.**
BTH screen for interactions between TssA1 and T6SS components of interest. A graphical representation (bottom) of β-galactosidase activity from *E. coli* DHM1 cells producing the indicated proteins fused to the adenylate cyclase T25 or T18 subunits and images of corresponding spots on X-gal LB agar plates (top) are shown. The combination of T25/T18 fusion proteins is indicated as 25 or 18 followed by the name of the fused T6SS protein. T6SS proteins are abbreviated as follows: A1 = TssA1, E1 = TssE1, F1 = TssF1, G1 = TssG1 and K1 = TssK1. The values shown on the *y*-axis correspond to the activity in Miller units. In each case, average activity from two independent experiments is shown and error bars indicate the standard deviation (SD). Experiments were carried out in triplicate.

**Figure 8.   Secondary-structure-weighted sequence alignment of TssK1 with gp8.**

A    Protein sequences were extracted from UniProt accession number BP08_BPT4 for enterobacteria phage T4 (gp8; bacteriophage T4) and the *Pseudomonas* Genome Database (TssK1; reference strain PAO1). Conserved positions are shown in black and grey background. The secondary structural elements corresponding to the 3D structure of gp8 (PDB code 1PDM) are shown above the alignment. Regions with significant sequence conservation are indicated with a corresponding block number.

B    Cartoon representation of the gp6/gp8 baseplate subcomplex of the T4 phage. The gp6 ring structure (PDB code 3H3W) is shown using the same colour code described in Fig 5. The chains A and B of the gp8 dimers (PDB code 1PDM) are shown in blue and yellow, respectively. The conserved blocks mapped onto the dimeric structure of gp8 are shown in orange. A close-up view of the gp8 structure (bottom) shows that the conserved blocks are involved in dimerization and interaction with gp6.

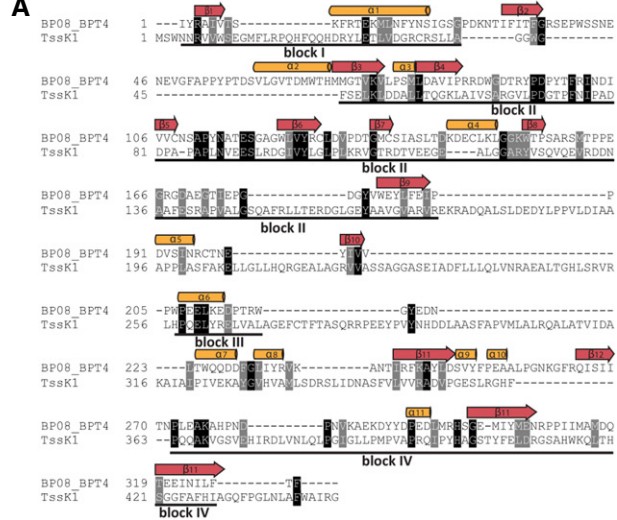

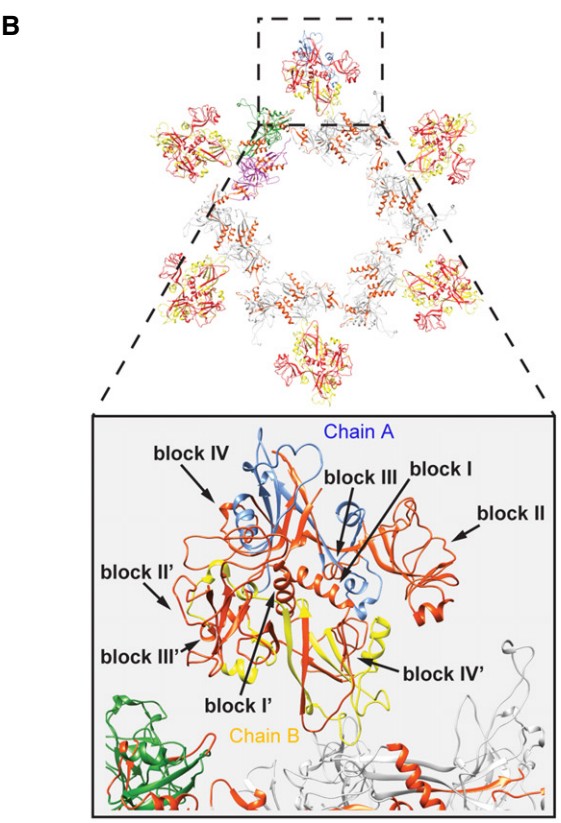

# Discussion

Major advances have been made regarding the structure and function of T6SS subassemblies, such as the sheath structure (Kube *et al*, 2014; Clemens *et al*, 2015; Kudryashev *et al*, 2015) and the inner membrane core complex (Durand *et al*, 2015). However, details on a T6SS putative baseplate, such as what is described in the T4 phage (Leiman *et al*, 2010) and R-type pyocin (Nakayama *et al*, 2000), remain scarce. In previous work, electron cryotomographs of *V. cholerae* cells (Basler *et al*, 2012) showed a bell-like structure connecting the T6SS tail sheath to the membrane, suggesting that a baseplate also exists in the T6SS. Here, we report the structure–function characterization of a putative *P. aeruginosa* T6SS baseplate subcomplex, TssA1.

Our AUC data and negative stain images indicate that TssA1 forms dodecameric ring-shaped structures with an outer diameter of ~260 Å and a central cavity of ~100 Å. We propose that the C-terminal region is responsible for oligomerization of the TssA1 ring complex and may form a rigid inner core, whereas the N-terminal domain may be flexible forming the outer region. We also show that TssA1 interacts at one extremity of the T6SS sheath structure and that the dimensions of the TssA1 ring are strikingly similar to the TssB1C1 sheath and Hcp1 hexamers (Mougous *et al*, 2006). Together, these results point to TssA1 as a possible component of a T6SS baseplate that would dock the TssBC sheath through which the Hcp tube passes. Remarkably, sequence and structural comparisons identified the C-terminal region of the gp6 phage baseplate component as a genuine homologue of TssA1. In phage, the C-terminal domain of gp6 forms a dodecameric ring crucial for proper phage tail assembly/contraction, with which several other baseplate components and the gp18 phage sheath (Aksyuk *et al*, 2009a) are connected (Fig 9).

In the three related structures, namely phage, R-type pyocin and T6SS (Leiman *et al*, 2010; Ge *et al*, 2015), the characterized shared components are tail tube-like (gp19 or Hcp), tail sheath-like (gp18 or TssBC) and tail spike-like (gp5-gp27 or VgrG). We show that TssA1 interacts with all of these components in the T6SS similar to gp6 in phage. As for the baseplate, phage and R-type pyocins share gp6, gp25 and gp53 (Kube & Wendler, 2015), of which only gp25 has been assigned a homologue in the T6SS, namely TssE (Bingle *et al*, 2008; Leiman *et al*, 2009; Lossi *et al*, 2011). In the T4 phage baseplate, the C-terminus of gp6 forms the interface for ring multimerization to provide a platform for the gp6 (N-terminal region), gp25 and gp53 subcomplex (Fig 9). These proteins interact with the first ring of the gp18 sheath. Here, we confirm that the gp6N-terminus has homology with TssF1 (Fig EV3A) and that TssA1 and TssF1 interact *in vivo*, suggesting that gp6 could be a fusion protein between these two T6SS components. Such chimeric proteins are not unique in terms of evolutionary relationships between phage and bacterial T6SS, since gp18 is a chimera between TssB and TssC (Aksyuk *et al*, 2009a), while VgrG is a fusion of gp27-gp5 (Leiman *et al*, 2009).

Given that the synteny in *tssEFG* organization in T6SS clusters is highly conserved, and TssE is a gp25 homologue, it is possible that TssG has a gp53-like function, which will recapitulate the overall organization of a T6SS baseplate complex as shown in our model (Fig 9). In support of this hypothesis, we observe sequence similarities between TssG1 and gp53 (6% identity and 40% similarity over 348 residues; Fig EV3B), which also confirm previously reported data with TssG from *E. coli* (Brunet *et al*, 2015). Although TssG is considerably longer than gp53, there is a conservation of predicted secondary structure for overlapping regions (Fig EV3B). In the outer radius of the phage, gp6 ring is another interacting subcomplex

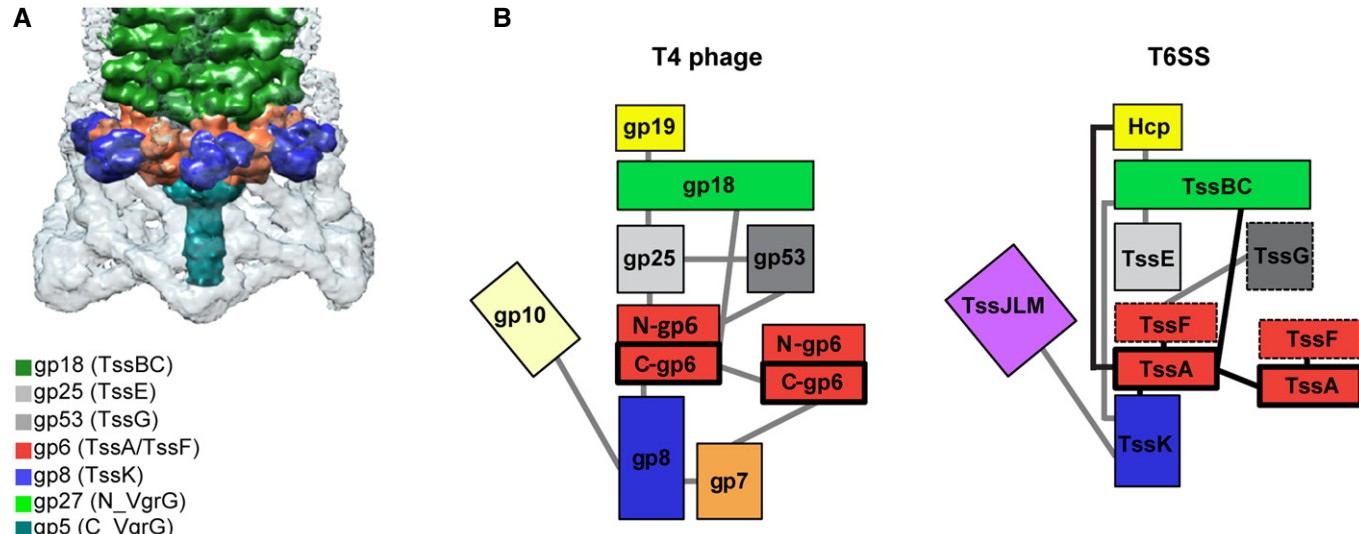

**Figure 9. Baseplate/tail organization in the T4 phage versus the T6SS.**

A  Structure of the sheath/baseplate complex of the bacteriophage T4, adapted from Kostyuchenko *et al* (2003). A colour code with corresponding gp protein is provided. The proposed corresponding homologue in the T6SS is indicated with parentheses.

B  The T4 bacteriophage baseplate organization versus a putative T6SS baseplate working model. Schematic representation (adapted from Aksyuk *et al*, 2009b) of the T4 bacteriophage baseplate/tail complex (left) and a putative similar T6SS structure (right). The protein–protein interaction network of both, T4 phage and T6SS is shown with grey lines. The interactions between T6SS proteins found in this work are shown with black lines.

    

formed by gp7 and gp8 (Fig 9). In phage, this part of the baseplate is also used to branch, via gp10 (Yap *et al*, 2016), the tail fibres that are unlikely to exist in the T6SS. Instead, using a variety of protein–protein interaction methods, we show that TssA1 interacts with TssK1, a membrane-associated protein (Casabona *et al*, 2013; English *et al*, 2014) that has been shown to interact with the T6SS membrane complex TssL/TssM (Zoued *et al*, 2013). Note that in phage, no such membrane complex is found, but as indicated above, the baseplate has to be connected to phage tail fibres. The TssA–TssK interaction is very consistent and has been found in other T6SSs such as in *E. coli* (Zoued *et al*, 2013). Interestingly, here we observe that TssK1 has similarity with gp8 (8% identity; 48% similarity; residues 37–444; Fig 8A), with several blocks of conservation (blocks I to IV). In gp8, some of these conserved blocks are involved in dimerization and interaction with gp6 (Fig 8B and Appendix Fig S1). Furthermore, the formation of a stable complex between TssK, TssF and TssG shown *in vitro* (English *et al*, 2014; Brunet *et al*, 2015) reinforces the connectivity between the different components of a putative T6SS baseplate and together these findings point to a structural parallel between the phage baseplate and a related complex in the T6SS (Fig. 9B).

Our study also reveals that TssA1 is not strictly essential for TssB1C1-sheath assembly. This is in contrast to the *tssE* mutant, in which the sheath does not assemble, as is also the case for *tssM* and *tssK* mutants (Basler *et al*, 2012; Kapitein *et al*, 2013; English *et al*, 2014; Gerc *et al*, 2015). This indicates that although TssA1 together with other proteins, including TssE1 and TssK1, may be part of a putative T6SS baseplate, these components exert distinct functions with TssA1 possibly acting downstream from TssK and TssE in the sheath assembly process. One possibility is that in the *tssA* mutant the sheath is assembled but mislocalized and/or degraded (Kostyuchenko *et al*, 2005; Basler *et al*, 2012; Zoued *et al*, 2013).

Recent studies have proposed that TssA acts to prime and polymerize the T6SS contractile sheath and is not part of the baseplate (Zoued *et al*, 2016). In this study, the TssA protein described is from *E. coli* (*Ec*TssA) which from our phylogenetic analysis belongs to the TssA2 family (Fig 1A). A careful examination of the sequences indicates that TssA1 and TssA2 proteins are very different with TssA2-like proteins having a C-terminal extension (Figs EV4 and EV5). The study on *Ec*TssA also reports the X-ray structure of the C-terminal domain which is proposed to form a double hexameric ring (PDB code 4YO5). This particular region is missing in TssA1 but instead appears to be structurally related to an accessory T6SS protein associated with the H1-T6SS, namely TagJ, which we previously showed interacts with the TssB1C1 sheath (Forster *et al*, 2014). In addition, while the gp6-like sequence homology blocks are well conserved in the TssA1 family of proteins, these are degenerated in the TssA2 family, as schematically represented in Fig EV5. It is thus clear that the TssA protein that we describe in our study, TssA1, and *Ec*TssA (or TssA2) are two distinct proteins and most of the hypothetical mechanism that has been associated with *Ec*TssA remains to be experimentally validated in other T6SSs. Furthermore, it is remarkable that we previously showed that there are at least two T6SS subclasses, which are based on the presence/absence of TagJ (Forster *et al*, 2014) and we could also show here that TssA1 interacts with TagJ (Fig EV4C).

In conclusion, our data suggest the existence of a phage-like baseplate component in the T6SS, namely TssA1, and describe a

stable baseplate/tail subcomplex for the T6SS. The existence in the T6SS of a gp6-like domain that we describe here within TssA1 is a breakthrough since gp6 is central to the phage baseplate (Kostyuchenko *et al*, 2005) organization. Finally, structure-based sequence homology and protein–protein interaction analyses clearly suggest the existence of a larger T6SS baseplate-like structure whose organization is proposed in this study although further structural studies are required to provide in-depth molecular details.

## Materials and Methods

### Bacterial strains, plasmids and culture conditions

All bacterial strains used in this study are listed in Appendix Table S1. *P. aeruginosa* strains were grown at 37°C in tryptone soy broth (TSB) or Luria-Bertani (LB) supplemented with appropriate antibiotics at the following concentration: streptomycin, 2,000 μg/ml; carbenicillin, 300 μg/ml; and tetracycline, 50–100 μg/ml. *E. coli* strains were grown in LB or terrific broth (TB) supplemented with appropriate antibiotics at the following concentration: kanamycin, 50 μg/ml; ampicillin, 50–100 μg/ml; streptomycin, 50 μg/ml; chloramphenicol, 34 μg/ml; and tetracycline, 15 μg/ml.

*Pseudomonas aeruginosa* in-frame deletion mutants were constructed using the suicide plasmid pKNG101 as described previously (Kaniga *et al*, 1991). Deletion of gene of interest was verified by PCR and the absence of the corresponding product verified by Western blot analysis. For the insertion of the *tssB1-sfGFP* gene fusion at the *P. aeruginosa* chromosomal *att* site, the Mini-CTX-plac-*tssB1-sfGFP* was conjugated in strains of interest by tri-parental mating and the presence of the *tssB1-sfGFP* gene confirmed by PCR.

All plasmids used in this study are listed in Appendix Table S2. To complement PAKΔ*retS*Δ*A1*, PAKΔ*retS*Δ*A1*Δ*B1*::*B1-sfGFP* and PAKΔ*retS*Δ*B1*Δ*E1*::*B1-sfGFP* strains, *tssA1* and *tssE1* genes were cloned into pBBR1MCS-4 (pBBR) vector yielding pBBR-*A1* and pBBR-*E1*, respectively. To produce His$_6$-TssA1 in *P. aeruginosa*, the corresponding DNA sequence was cloned into pBBR1MCS-4 yielding pBBR-*His$_6$A1* which encodes an N-terminally His$_6$-tagged TssA1. The pBBR derivative plasmids were introduced in *P. aeruginosa* strains by electroporation.

For production of His$_6$-TssA1 and His$_6$-TssA1$_{1–245}$ in *E. coli* B834 (DE3), the *tssA1* gene and the sequence corresponding to the first 245 residues of TssA1 were amplified from *P. aeruginosa* genomic DNA and cloned into pET28a. The resulting recombinant plasmids pET-*A1* and pET-*A1$_{1–245}$* encode for TssA1 and TssA1$_{1–245}$, respectively, both with a N-terminal cleavable His$_6$ tag. For co-purification of His$_6$-TssA1 and TssB1C1, *tssA1* and *tssB1C1* genes were amplified from genomic DNA of *P. aeruginosa* and cloned into pACYC-duet (pACYC) vector to insert *tssA1* into the first multiple cloning site (MCS1) and *tssB1C1* into the second multiple cloning site (MCS2). The resulting recombinant pACYC-*A1-B1C1* plasmid encodes TssA1 with an N-terminal His$_6$ tag together with untagged TssB1C1.

For BTH analysis, the gene of interest was amplified from *P. aeruginosa* genomic DNA and cloned into plasmid pKT25 or pUT18C leading to in-frame fusions of the protein of interest with

the T25 or T18 subunit of the *Bordetella pertussis* adenylate cyclase, respectively.

All constructs were confirmed by sequencing (GATC Biotech, Germany) prior to use.

### Preparation of supernatant from *P. aeruginosa* cultures and Western blotting

*Pseudomonas aeruginosa* strains were grown overnight in TSB and subcultured to an $OD_{600}$ of 0.1 until early stationary phase at 37°C. Cells were separated from culture supernatants by centrifugation at 4,000 *g* at 4°C and resuspended in 1× Laemmli buffer. Proteins from the culture supernatant were precipitated with 10% TCA, washed in 90% acetone, air-dried and resuspended in 1× Laemmli buffer for analysis.

Protein samples or cell extract were migrated onto SDS–PAGE and transferred to a nitrocellulose membrane at 3 mA/cm². After transfer, membranes were blocked overnight in blocking buffer (5% milk powder, 0.1% Tween-20 in Tris-buffered saline, pH 8.0). Polyclonal antibodies against TssA1, Hcp1, Tse3 and VgrG1a were used at a dilution of 1:1,000. Monoclonal anti-His₆ and anti-RNA polymerase antibodies were used at a dilution of 1:5,000. Secondary antibodies conjugated to horseradish peroxidase were used at a dilution of 1:5,000. Western blots were developed using Super-Signal West Pico Chemiluminescent Substrate (Pierce) and visualized on a LAS3000 Fuji Imager.

### Protein purification

Expression vectors pET-*A1*, pET-*A1₁₋₂₄₅* and pACYC-*A1-B1C1* were transformed into *E. coli* B834 (DE3) cells for production and purification of recombinant proteins. Cells were grown at 37°C to an $OD_{600}$ of about 0.6 in TB. Expression was subsequently induced using 0.5 mM isopropyl β-D-thiogalactoside (IPTG) and cells were grown overnight at 18°C before centrifugation (4,000 *g*, 15 min at 4°C).

Cells were resuspended in buffer A (25 mM Tris–HCl pH 7.5, 150 mM NaCl, 10 mM imidazole), lysed by sonication after the addition of an anti-protease cocktail (Sigma), and cell debris were eliminated by centrifugation (18,000 *g*, 50 min). Proteins were purified by IMAC chromatography using Ni-NTA resin (Qiagen) equilibrated in buffer A. Proteins were washed in the affinity buffer supplemented with 20 mM imidazole and eluted within the same buffer supplemented with 300 mM imidazole. Eluted fractions were pooled and loaded onto a Superdex 200 10/300 column (GE Healthcare) column equilibrated in 25 mM Tris–HCl pH 7.5 and 150 mM NaCl. All chromatographic steps were performed on an ÄKTA prime plus (GE Healthcare) at 4°C. Protein purity was checked by SDS–PAGE.

### *In vitro* cross-linking and multiangle laser light scattering (SEC-MALS)

The His₆-TssA1 and His₆-TssA1₁₋₂₄₅ proteins were purified in 50 mM NaH₂PO₄ pH 7.5, 150 mM NaCl as described above. About 30 μg of protein was incubated with increasing concentration of ethylene glycol-bis(succinimidylsuccinate) (EGS) (2 and 5 mM) and samples incubated at room temperature during 30 min, 1 or 2 h. The reaction was stopped by the addition of 25 mM Tris (pH 8.0) (final concentration). Samples were analysed on 3–12% gradient

SDS–PAGE and cross-linked species were identified by Western blotting using anti-His₆ antibodies as described above.

SEC-MALS experiments were performed at room temperature by loading the purified His₆-TssA1₁₋₂₄₅ protein on a Superdex 200 10/300 column (GE Healthcare) or the full-length His₆-TssA1 on a WYATT 100S5 column (50,000–7,500,000 Da separation range), coupled to WYATT MALS instruments (Wyatt Technology Corp.). Samples were eluted in buffer containing 25 mM Tris–HCl pH 7.5 and 150 mM NaCl and data were processed using the ASTRA 5.3.4 software.

### Analytical ultracentrifugation

Analytical ultracentrifugation (AUC) experiments were obtained on two Beckman XL-I instruments equipped with AnTi50 or AnTi60 rotor using two-sector cells with column heights of 12 mm at 20°C. Sedimentation velocity experiment data were acquired for the purified His₆-TssA1 (25 mM Tris–HCl pH 7.5, 200 mM NaCl) at concentrations of 0.8, 0.56 and 0.32 mg/ml at rotor speeds of 20,000 and 30,000 rpm and for the purified TssA1₁₋₂₄₅ (25 mM Tris–HCl pH 7.5, 200 mM NaCl) at concentrations of 1.6, 1.2 and 0.8 mg/ml at rotor speed of 50,000 rpm. SEDFIT software (version 14.6) was used to analyse the absorbance data recorded at 280 nm. The Lamm equation was directly fitted to every third boundary scan in order to obtain the size-distribution analysis c(s) that provided the $S_{20,w}$ and mass values. The final c(s) analyses were based on a fixed resolution of 200 and optimized by floating the meniscus, the bottom of the cell, the baseline and the average frictional ratio f/fo until the RMSDs and appearance of the fits were appropriate.

### Pull-down assay and MS analysis

His₆-TssA1 was produced from a *tssA1* mutant carrying the pBBR-*His₆-A1*. The experimental control is a strain carrying the empty pBBR vector. Cells were grown at 37°C in 500 ml TSB supplemented with carbenicillin to an $OD_{600}$ of about 5, resuspended in buffer A (25 mM Tris–HCl pH 7.5, 150 mM NaCl, 10 mM imidazole) and lysed by sonication at 4°C and cell debris were eliminated by centrifugation (18,000 *g*, 50 min). Samples were loaded onto Ni-NTA resin column (Qiagen) equilibrated in buffer A, washed and eluted in buffer B (25 mM Tris–HCl pH 7.5, 150 mM NaCl, 300 mM imidazole). Eluted fraction were pulled and used immediately for MS analysis.

LC-MS/MSe experiments were performed on an ESI quadrupole-time-of-flight (Q-TOF) instrument (SYNAPT G2-S, Waters Ltd.) equipped with a M-Class nanoAcquity 2D-LC system. Samples were trapped on a nanoAcquity Symmetry C18, 5 μm trap column (180 μm × 20 mm, Waters) and separated on a nanoAcquity HSS T3 1.8 μm C18 capillary column (75 μm × 250 mm, Waters). The gradient used was 3% MeCN in 0.1% formic acid (aq.) to 50% MeCN in 0.1% formic acid (aq.) over 65 min at a flow rate of 0.3 μl/min with an analytical column temperature of 50°C. Approximately 50 ng of digested material was injected. Positive ion mass spectrometric data were acquired in data-independent MSe/resolution mode, with the scan time set to 0.75 s. Data acquisition was performed using MassLynx 4.1 software (Waters, Manchester). Instrument settings were as follows: acquisition range 50:3,000 Da, capillary voltage 3.0 kV, sample cone 40 V, source temperature

80°C, cone gas 100 l/h, nano-flow gas 0.4 bar, purge gas 50 l/h. Low-energy function trap collision energy was set to 4 V, while the high-energy function collision energy was ramped from 14 to 60 V. Glu-fibrinopeptide B and Leu-enkephalin were used as lockmass ions for the analyses. The resulting data were searched against the *P. aeruginosa* PAO1 genome from the *Pseudomonas* Genome DB resource using ProteinLynx Global Server software v3.0.1 (Waters, Manchester). Search criteria were as follows: peptide tolerance 10 ppm, fragment ion tolerance 0.1 Da, minimum fragment ion matches per peptide 3, minimum fragment ion matches per protein 7, minimum peptide matches per protein 1, false discovery rate 4, digest reagent trypsin (without proline exclusion). Fixed modifier—carboxymethyl (Cys), variable modifier—oxidation (Met).

### Nanogold labelling

Size-exclusion chromatography eluted fractions containing co-purified His$_6$-TssA1 and TssB1C1 were incubated on ice for 1 h min with Ni-NTA nanogold (Nanoprobes Inc.). Immunogold-labelled sample was subsequently loaded onto a Superdex 200 10/300 column (GE Healthcare) equilibrated in 25 mM Tris–HCl pH 7.5 and 150 mM NaCl in order to remove excess or unbound gold. SEC eluted fractions were used immediately for EM sample preparation.

### Negative stain electron microscopy and image processing

Two microlitre of purified TssA1 or immunolabelled His$_6$-TssA1 with TssB1C1 was applied to a glow-discharged copper mesh EM grid overlaid with a continuous carbon support film. Samples were then negatively stained with 2% (w/v) uranyl acetate and imaged using an FEI Tecnai 120 kV electron microscope. For TssA1 single-particle analysis, sets of images were collected at a magnification of 67,000× using a defocus range of 1–2 μm. Charge-coupled device (CCD) images were recorded on a 2,048 × 2,048 pixel TemCam-F216 camera (TVIPS) using a dosage of ~45 e-/Å$^2$. The final pixel size of the resulting images was 2 Å/pixel. Image defocus and contrast transfer function (CTF) determination was performed using CTFFIND3 (Mindell & Grigorieff, 2003). Images free of drift and astigmatism were CTF-corrected by phase flipping using the command bctf in Bsoft (Heymann & Belnap, 2007). A total of 3,714 isolated TssA1 complex particles were selected in a semi-automated manner using the EMAN suite of programs. At this point, analysis of the data set was performed independently using IMAGIC-5 and, in parallel, RELION (REgularized LIkelihood OptimizatioN). The single-particle approach was used to analyse the images in each case. In IMAGIC-5, all images were band-pass-filtered, to remove uneven background and high-frequency noise, and normalized, to ensure all particles have the same scale of densities. After reference-free alignment to centre the particles, TssA1 complex images were classified by multivariate statistical analysis (MSA). Successive iterations of reference-free alignments and MSA classification led to class averages displaying the TssA1 complex as lobed rings. The double MSA method (Elad *et al*, 2008) was carried out by classifying initially aligned images of the data set into 12 classes based on the first 9 eigenimages that represent variable orientations. After extracting all members belonging to each orientation class into separate groups, subclassification into 3 further classes was performed on the basis of 2–4 eigenimages that showed local variations.

Difference images were then calculated by subtracting the orientation class averages from each of their corresponding 3 subclass averages. In RELION, following pre-processing (normalization and masking), the data set was subjected to a Bayesian method of reference-free two-dimensional (2D) class averaging. In total, three rounds of classification were performed until convergence was observed between consecutive class averaging. Twenty-five iterations were executed per round, while ensuring particles from the best class averages obtained in the previous round were selected for use in the next one. Additional optimization of the procedure involved increasing the regularization parameter (T; from 1 to 2) and in-plane angular sampling (5 to 3°) for the final round of class averaging to further refine particle image alignment and classification. The last classification of the sorted data divided into 60 classes (30 particles per class) and twelve representative classes are shown.

### Bacterial two-hybrid and β-galactosidase assays

*In vivo* protein–protein interactions were analysed using the bacterial two-hybrid (BTH) system (Karimova *et al*, 1998) as detailed previously (Forster *et al*, 2014). For quantification of BTH interactions, β-galactosidase activity from co-transformants picked from X-gal LB agar plates was measured (Miller units) as previously described (Miller, 1992). The experiments were done at least in triplicate and a representative result is shown.

### Fluorescence microscopy

Overnight cultures of *P. aeruginosa* PAKΔretSΔ*tssB1::B1-sfGFP* derivatives strains were diluted into fresh LB broth and cultivated to OD$_{600}$ of about 0.8–1. Cells were washed and resuspended in PBS 1× (to reach 10 OD units). 1 μl of washed cells was spotted on thin pad of 1% agarose in PBS covered with a coverslip and immediately imaged at room temperature using an objective heated to 37°C. Coverslips were mounted and analysed using a ZEISS Axio Imager fluorescence microscope. Image analysis has been performed using Fiji (Schindelin *et al*, 2012).

### Bioinformatic analysis

*Pseudomonas aeruginosa* protein sequences have been retrieved from the Pseudomonas Genome Database (http://www.pseudomonas.com). For phylogenetic analysis, protein sequences were retrieved by BLASTP search using *P. aeruginosa* TssA1, TssA2 and TssA3 as query against the Kyoto Encyclopedia of Genes and Genomes database (KEGG) (http://www.genome.jp/tools/blast/). The sequence of EcTssA (GenBank accession number 284924261) was added further as this sequence could not be retrieved from the KEGG. TssA sequences were aligned with MAFFT and likelihood-based phylogenetic analysis was conducted using MEGA version 6 (Tamura *et al*, 2013). The secondary structure prediction of TssA1 shown in Fig 3C has been performed using the web-based prediction server ESPript 3 (http://esprit.ibcp.fr). The secondary-structure-weighted sequence alignment of TssA1 with the C-terminal region of gp6, TssF1 with the N-terminal moiety of gp6, TssK1 with the gp8 protein and TssG1 with gp53 was carried out with PROMALS3D. Mapping of sequence conservation into the 3D

structures of gp6 (PDB codes 3H2T, 3H3W and 3H3Y) and gp8 (PDB code 1PDM) was carried out with UCSF Chimera. Structural model was predicted using Phyre2 (Kelley *et al*, 2015).

**Expanded View** for this article is available online.

## Acknowledgements

We thank J. Gor and S. J. Perkins for use of the UCL Molecular Interactions Facility for the AUC measurements, S. J. North and P. Hitchen for use of the CISBIO Mass Spectrometry Core Facility at Imperial College London, N. Lossi for initiating the work and A. Förster for help with the phylogenetic tree. Diamond Light Source (Didcot, UK) and R. Rambo are acknowledged for access to their HPLC-MALS instrument (proposal SM14619-1), with financial support from Biostruct-X (project 10774). This work was supported by the MRC programme grant MRK/K001930/1.

## Author contributions

AF, PSF, SP, DAJ and OS conceived and designed experiments, and wrote the paper. SP, OS and EM conducted experiments.

## Conflict of interest

The authors declare that they have no conflict of interest.

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
