## [Review Process File · The EMBO Journal]

Manuscript EMBO-2016-94024

TssA forms a gp6-like ring attached to the type VI secretion sheath

Sara Planamente, Osman Salih, Eleni Manoli, David Albesa-Jové, Paul Freemont and Alain Filloux

Corresponding author: Alain Filloux, Imperial College London

Review timeline:

Submission date:	3 February 2016
Editorial Decision:	15 February 2016
Revision Received:	06 April 2016
Editorial Decision:	13 May 2016
Revision received:	20 May 2016
Accepted:	23 May 2016

Editor: David del Alamo

Transaction Report:

Please note that the manuscript was transferred from another journal where it was originally reviewed. Since the original reviews are not subject to EMBO's transparent review process policy, the reports and author response cannot be published.

1st Editorial Decision

15 February 2016

Thank you for the submission of your manuscript entitled "A gp6 ring-like structure at the tip of the type VI secretion sheath" and please accept my apologies for the delay in getting back to you. We have now received the comments from the arbitrating referee, which I copy below.

As you can see from his/her comments, the referee is rather supportive of your work although s/he believes that to some extent the conclusions are not fully supported by the evidence presented and need to be toned down. S/he also points out to a number of concerns that will require your attention before your manuscript can be published in The EMBO Journal. I will not repeat here these issues, which I believe are rather clear, but please notice that they may require additional experimental evidence in at least some cases.

Arbitrating Referee Report

I think the answer is that the paper could be interesting IF they will do more to address the referees' concerns. In general the figures are relatively poor and could do with a lot of tidying.

In general I think they need to be willing to tone down the language in the paper to acknowledge the fact that their data may suggest lots of things but don't really provide conclusive evidence for much at this point in time... additionally the experiments the referees suggest that I think are reasonable for them to do better are:

1. Mass Determination - Ref#1 points out that they seem to have used the wrong AUC experiment for shape

independent mass determination. They do not address this in their response and they also give no error estimate on the mass. Their explanation for why they can't do SEC-MALS is poor - if the sample is really a mix of dodecamers and aggregates of this it should be entirely possible to find a column to separate these species and then get a mass from SEC-MALS

On a related point - their evidence that the C-terminally deleted protein is monomeric is accepted by the two referees but looks rather dodgy to my eyes - the mass estimate is sloped and suggests that this construct is aggregating (whether in a biologically relevant way or simply because the protein is unhappy cannot be distinguished). The claim that they know which portion of the protein is driving oligomerisation is therefore overstated.

2. EM - neither of the referees particularly comment on the EM but I think there are some issues with this. Again it seems rather over interpreted given the images presented (although I haven't seen the supplement where further analysis is reported). The level of agreement between the re-projection of the model and the images is suggestive but not conclusive.

3. Which end - I suspect rather than doing further fluorescence experiments it would be better just to tone down the language surrounding this.

4. Ref#4 describes the fluorescence data as not compelling - I would tend to agree and think that the authors should address these concerns if they want to present the fluorescence data in the paper.

1st Revision – author's response

15 February 2016

ANSWER TO REVIEWER'S COMMENTS

We have now prepared a revised version of our manuscript entitled “TssA forms a gp6-like ring attached to the type VI secretion sheath” which replaces the previous title “A gp6 ring-like structure at the tip of the type VI secretion sheath”.

We have carefully considered the comments of the arbitrating referee, and you will find below a point-by-point answers to these comments with our answers written in blue. The comments are mainly querying biochemical and EM approaches which are now described in more detail while some have been added or repeated.

In addition to address the referee comments, we have included in the discussion a paragraph which comments on the recent paper by Zoued et al., published in Nature where the structure and fate of a TssA protein from the Escherichia coli T6SS (EcTssA) is described (Zoued et al., Nature, 2016, 531(7592):59-63).

What is noticeable is that this particular EcTssA is remarkably different from the one we studied here (PaTssA1) and our phylogenetic analysis (added in Figure 1 of the revised version of our paper), shows that they belong to distinct subclasses. Moreover, we found that the structure of the C-terminal domain of EcTssA (PDB code 4YO5) that they report, is missing in our TssA1 but shows instead structural similarities to TagJ, an accessory component of the P. aeruginosa T6SS-H1, (Fig EV4 and EV5 in our manuscript). We also provide evidence that TagJ interacts with PaTssA1, which is in agreement with the newly identified “TagJ-like extension” found in EcTssA and absent in PaTssA1 (Fig EV4C).

Overall the discrepancy between the two studies matches very nicely with the two subclasses of T6SS that we have previously reported (Förster et al., JBC, 2014, 289:33032-33043), one using the TssA1/TagJ system whereas the other is using a TssA from the EcTssA family. Most importantly the Zoued et al. study fails to report the homology with the gp6 C terminus which is the core piece of information in our paper, together with the fact that TssK is a gp8 homologue. From this angle we are confident that our study is fully original and distinct/complementary from the Zoued et al. paper.

1. Mass Determination - Ref#1 points out that they seem to have used the wrong AUC experiment for shape independent mass determination. They do not address this in

their response and they also give no error estimate on the mass.

We believe that we used appropriate biochemical methods and give here below the main reasons.

Since the previous version was written for Nature, the details on the experimental approaches may have been too scarce to make clear what we have actually been doing which is also fixed in the new materials and methods written for the EMBO J version. In brief, sedimentation velocity AUC experiments have given accurate molecular weight values for protein complexes, including ring shaped complexes, as shown in the following peer reviewed articles:

(Okemefuna et al., 2009, J Mol Biol 91(1):119-35; Niewiarowski et al., 2010, Biochem J, 429(1):113-25; Ando et al. 2011, PNAS, 108(52):21046-51).

A notable advantage of the velocity sedimentation is that it usually requires only several hours over the sedimentation equilibrium (several days), thus sedimentation velocity can be used with samples that are unstable. To avoid problems due to the lack of the stability of TssA1 complex over the course of the experiment (carried out at 20°C) we therefore used this AUC method to estimate the molar mass of TssA1 in solution.

A detailed description of the AUC experiments performed in this study is provided below.

Two independent preparations of the purified TssA1 have been analysed by sedimentation velocity experiments at different rotor speed (20,000 rpm and 30,000 rpm).

Figure A. SDS-PAGE analysis of TssA1 after purification steps

The purity of the sample is shown by the SDS-PAGE in Figure A.

The sedimentation coefficient distribution function $c(s)$ was calculated by direct fitting of the sedimentation boundaries (use of SEDFIT software) for all data sets. Good fits to the sedimentation boundaries were obtained in all cases and a representative fit is shown in Figure 2C of our new version. One of the parameters used to optimize the $c(s)$ fit, is the frictional ratio (f/f_0) from which shape information can be inferred. During $c(s)$ fit optimization the frictional ratio has been floated till the RMSD values and the fits were suitable. The error estimate on the mass of the peaks observed in all the size-distribution analyses $c(s)$ is provided in the revised version of the manuscript.

Their explanation for why they can't do SEC-MALS is poor - if the sample is really a mix of dodecamers and aggregates of this it should be entirely possible to find a column to separate these species and then get a mass from SEC-MALS.

We agree with the concern of the arbitrating referee. In an attempt to separate TssA1 complexes from the aggregates, an optimal SEC column (the WYATT 100S5 column with 50,000–7,500,000 Da separation range) has been used coupled to a MALS instrument in order to get a good separation of the TssA1 complexes from aggregates. Adoption of this new condition allowed separation of TssA1 complexes from large aggregates as shown in Figure EV1A of the revised version. The SEC profile indicates TssA1 is polydisperse in solution.

SEC-MALS data indicate that the first eluted peak (elution volume of about 10 ml) has a molar mass of 1.0 ± 0.1 MDa which is in agreement with the AUC data where a minor peak at 28.3–29 S with mass estimate of 1.05 ± 0.05 MDa is also observed. The SEC profile shows additional peaks between elution volume range 12 to 14 ml, unfortunately, due to peak overlap, the accurate molar masses of these peaks could not be assessed. In conclusion, AUC remains the most reliable technique to estimate the molecular mass of TssA1 in solution as it allowed detection of the different species present in the sample from the peaks in the size distribution analyses $c(s)$. Moreover, sedimentation velocity experiments permit distinction of the most abundant TssA1 species present in solution, indicating that TssA1 is mainly a dodecamer.

On a related point - their evidence that the C-terminally deleted protein is monomeric is accepted by the two referees but looks rather dodgy to my eyes - the mass estimate is sloped and suggests that this construct is aggregating (whether in a biologically relevant way or simply because the protein is unhappy cannot be distinguished). The claim that they know which portion of the protein is driving oligomerisation is therefore overstated.

SEC-MALS experiments using the truncated TssA1₁₋₂₄₅ protein have been repeated using optimal concentration of the protein and best experimental conditions. The SEC-MALS data (Figure 3D) showed the TssA1₁₋₂₄₅ elution peak with a molar mass of 28.4 ± 1.06 kDa which correspond to a monomer (theoretical MW of the protein is about 27 kDa). The MALS plot across the UV trace of the peak does not appear sloped in the latest experiment. Additionally, AUC experiments have been carried out using the purified truncated protein and confirm the monomeric state of TssA1₁₋₂₄₅ (Figure EV1 C). SEC-MALS (Figure 3D), in vitro cross-linking and AUC (Figure EV1, panels B and C) data showed that the truncated TssA1₁₋₂₄₅ protein is a monomer which strongly suggest that the C-terminal region of TssA1 is involved in the oligomerisation.

2. EM - neither of the referees particularly comment on the EM but I think there are some issues with this. Again it seems rather over interpreted given the images presented (although I haven't seen the supplement where further analysis is reported). The level of agreement between the re-projection of the model and the images is suggestive but not conclusive.

Additional information/description of the TssA1 single-particle analysis has been added in the latest version of the manuscript (Figure 3A and B and Figure EV2 in the new version of our manuscript).

We now provide a new 2D classification carried out using RELION and 12 representative classes of the TssA1 complexes are shown in Figure 3A, right panel. This clearly shows that TssA1 complexes are distinct rings which exhibit profound conformational heterogeneity and display a variety of symmetries ranging from three-fold to six-fold like symmetry. TssA1 complexes assume preferentially top-view orientations with very few side-views. Representative classes of TssA1 sideviews are shown: the Class 7 in Figure 3A (RELION classification), Classes 5 and 10 in Figure EV2A (IMAGIC-5 classification).

Additionally, a double MSA classification of TssA1 particles has been performed and shown in Figure 3B. This analysis highlights changes in external lobe position (Figure 3B, right panel, coloured circles), which is in agreement with our hypothesis that the TssA1 ring complex is highly flexible, although it is difficult to completely eliminate in-plane rotational alignment heterogeneity.

We agree that comparisons between reprojections of gp6_C rings and EM images of TssA1 rings are ultimately suggestive rather than conclusive. However, we undeniably observe similarities in terms of shape and size between the TssA1 and gp6_C rings that are clearly demonstrated in Figure EV2 of our new version. TssA1 rings have diameter, width and central hole dimensions of ~260 Å, ~95 Å and ~100 Å, respectively, and are clearly comparable to those measured for gp6_C rings at ~240 Å, ~80 Å and ~100 Å, respectively. These observations, together with significant sequence similarity (~62 %) detected between the C-terminal regions of TssA1 and gp6, hint at structural parallels existing between these two ring complexes.

3. Which end - I suspect rather than doing further fluorescence experiments it would be better just to tone down the language surrounding this. 4. Ref#4 describes the fluorescence data as not compelling - I would tend to agree and think that the authors should address these concerns if they want to present the fluorescence data in the paper.

We agree with the referee and we have toned down the interpretation and discussion of the fluorescence microscopy data which are now kept to a minimum and simply used to show the difference in impact of TssA when compared to other T6SS components. In brief, only clear observations showing the presence of TssB1 foci being decreased in the tssA1 mutant population compare to the WT cells, are described. Hypothesis regarding sheath dynamics in the observed strains has been removed as suggested by the referee.

2nd Editorial Decision

13 May 2016

Thank you for the submission of your revised manuscript to The EMBO Journal. As you will see below, your article was sent back to the original referee, who as I already mentioned to you, now considers that you have properly dealt with the main concerns originally raised in the review process, and therefore I am writing with an 'accept in principle' decision. This means that I will be happy to formally accept your manuscript for publication once a few more minor issues have been addressed.

Browsing through the manuscript myself I have noticed a few cosmetic issues that will need to be addressed in the final version of the paper. Essentially, figures involving BTH beta-gal measurements need to be described in better detail. In particular, error bars shown in figures 2A, 4D and 7 need to be defined in the figure legend. Although also mentioned in the methods section, it would also be helpful to include information regarding number of experiments performed.

REFeree REPORTS

Referee #1:

The authors have added significant new data that, whilst not changing their earlier interpretations, substantially strengthen the data presented in support of their ideas. This, taken together with the textual alterations, make the manuscript suitable for publications.

2nd Revision - authors' response

20 May 2016

Many thanks for your feedback on the manuscript and we are all extremely pleased with the accept decision.

I have also followed your recommendation as for the cosmetic issues you mentioned in your letter and this has all been fixed accordingly in the revised version I have uploaded.

3rd Editorial Decision

23 May 2016

I am pleased to inform you that your manuscript has been accepted for publication in the EMBO Journal.

Corresponding Author Name:

Journal Submitted to:

Manuscript Number: